# Rising Trend in the Prevalence of HPV-Driven Oropharyngeal Squamous Cell Carcinoma during 2000–2022 in Northeastern Italy: Implication for Using p16^INK4a^ as a Surrogate Marker for HPV-Driven Carcinogenesis

**DOI:** 10.3390/cancers15092643

**Published:** 2023-05-07

**Authors:** Paolo Boscolo-Rizzo, Jerry Polesel, Annarosa Del Mistro, Elisabetta Fratta, Chiara Lazzarin, Anna Menegaldo, Valentina Lupato, Giuseppe Fanetti, Fabrizio Zanconati, Maria Guido, Vittorio Giacomarra, Enzo Emanuelli, Margherita Tofanelli, Giancarlo Tirelli

**Affiliations:** 1Department of Medical, Surgical and Health Sciences, Section of Otolaryngology, University of Trieste, Strada di Fiume 447, 34149 Trieste, Italy; paolo.boscolorizzo@units.it (P.B.-R.); mtofanelli@units.it (M.T.); tirellig@units.it (G.T.); 2Unit of Cancer Epidemiology, Centro di Riferimento Oncologico di Aviano (CRO) IRCCS, Via F. Gallini 2, 33081 Aviano, Italy; 3Immunology and Molecular Oncology Diagnostics, Veneto Institute of Oncology IOV-IRCCS, Via Gattamelata 64, 35128 Padova, Italy; annarosa.delmistro@iov.veneto.it; 4Unit of Immunopathology and Cancer Biomarkers, Department of Translational Research, Centro di Riferimento Oncologico di Aviano (CRO) IRCCS, Via F. Gallini 2, 33081 Aviano, Italy; efratta@cro.it; 5Unit of Otolaryngology, AULSS 2 Marca Trevigiana, Piazzale dell’Ospedale 1, 31100 Treviso, Italy; anna.menegaldo@hotmail.it (A.M.); enzoemanuelli@libero.it (E.E.); 6Unit of Otolaryngology, General Hospital “S. Maria degli Angeli”, Via Montereale 24, 33170 Pordenone, Italy; valentina.lupato@asfo.sanita.fvg.it (V.L.); vittorio.giacomarra@asfo.sanita.fvg.it (V.G.); 7Department of Radiation Oncology, Centro di Riferimento Oncologico di Aviano (CRO) IRCCS, Via F. Gallini 2, 33081 Aviano, Italy; giuseppe.fanetti@cro.it; 8Department of Medical, Surgical and Health Sciences, Section of Pathology, University of Trieste, Strada di Fiume 447, 34149 Trieste, Italy; fabrizio.zanconati@asugi.sanita.fvg.it; 9Department of Medicine, Section of Pathology, University of Padova, via Giustiniani 2, 35128 Padova, Italy; mguido@unipd.it

**Keywords:** biomarker, epidemiology, head and neck cancer, human papillomavirus, oropharyngeal cancer, p16^INK4a^, prevalence, squamous cell carcinoma

## Abstract

**Simple Summary:**

The prevalence of oropharyngeal squamous cell carcinomas (OPSCCs) driven by human papillomavirus (HPV) infection has a wide geographical variability. In addition, several authors have reported steadily increasing prevalence rates over the past two decades. It is important to know the epidemiological landscape of these tumors both to estimate the reliability of diagnostic tests and to guide public health choices aimed at primary prevention. In this retrospective study, we observed a significant increase in the prevalence of HPV-driven OPSCC from 12% during 2000–2006 to 50% during 2019–2022. The use of p16^INK4a^ overexpression as a surrogate marker of transforming HPV infection should consider the prevalence rates of HPV-driven OPSCC as this significantly impacts on its positive predictive value. Health policies should actively promote vaccination against HPV in both females and males also for the prevention of OPSCC.

**Abstract:**

Background: The prevalence and incidence of oropharyngeal squamous cell carcinomas (OPSCCs) driven by human papillomavirus (HPV) infection are increasing worldwide, being higher in high-income countries. However, data from Italy are scanty. p16^INK4a^ overexpression is the standard in determining HPV-driven carcinogenesis, but disease prevalence impacts on its positive predictive value. Methods: This is a multicenter retrospective study enrolling 390 consecutive patients aged ≥18 years, diagnosed with pathologically confirmed OPSCC in Northeastern Italy between 2000 and 2022. High-risk HPV-DNA and p16^INK4a^ status were retrieved from medical records or evaluated in formalin-fixed paraffin-embedded specimens. A tumor was defined as HPV-driven when double positive for high-risk HPV-DNA and p16^INK4a^ overexpression. Results: Overall, 125 cases (32%) were HPV-driven, with a significant upward temporal trend from 12% in 2000–2006 to 50% in 2019–2022. The prevalence of HPV-driven cancer of the tonsil and base of the tongue increased up to 59%, whereas it remained below 10% in other subsites. Consequently, the p16^INK4a^ positive predictive value was 89% for the former and 29% for the latter. Conclusions: The prevalence of HPV-driven OPSCC continued to increase, even in the most recent period. When using p16^INK4a^ overexpression as a surrogate marker of transforming HPV infection, each institution should consider the subsite-specific prevalence rates of HPV-driven OPSCC as these significantly impact on its positive predictive value.

## 1. Introduction

The prevalence of oropharyngeal squamous cell carcinoma (OPSCC) varies mainly by geographic area, with it being much higher in Northern Europe and Northern America, and lower in Southern Europe and in low-income countries [1]. However, there was a fairly consistent increase in the fraction of these tumors attributable to human papillomavirus (HPV) infection over the most recent time periods [2]. 

HPV-driven OPSCC represents a separate entity characterized by a distinct genetic profile [3], different tumor immune microenvironment [4], and a higher chemo- and radio-sensitivity [5,6], with this resulting in a significantly higher overall survival compared with HPV-negative counterparts [7,8]. Thus, beside the epidemiological value, determining whether OPSCC is the consequence of a transforming infection by high-risk HPV strains or not has important clinical implications. In the clinical setting, HPV-driven OPSCC is determined mainly on the basis of p16^INK4a^ immunohistochemical (IHC) positivity, which is considered a surrogate for HPV-driven cancer. However, an up-regulation of p16^INK4a^ was identified in 8–20% of tumors with no evidence of HPV transforming infection [9,10], pointing the attention on the reliability of p16^INK4a^ as a stand-alone test for the identification of HPV-driven OPSCC [11]. The VIII edition of the TNM Staging System of Head and Neck Tumors deals separately with the staging of p16^INK4a^ positive and p16^INK4a^ negative oropharyngeal tumors in order to improve prognostic stratification, as different aspects of the disease, including category T and N, and the presence of extracapsular lymph node extension, have a different prognostic impact in the two groups [12].

Numerous trials are also underway to evaluate the non-inferiority of de-intensified treatments in HPV-driven OPSCCs whose rationale rests on the observation that HPV-driven OPSCC has a much better prognosis and affect younger patients [13]. Thus, these trials aim to identify treatments that reduce toxicity while maintaining efficacy in order to improve long-term quality of life. Therefore, in the near future, an accurate diagnosis of HPV-driven OPSCC might have even more significant implications and guide the clinician towards specific therapeutic strategies. 

A very comprehensive systematic review and meta-analysis estimated a sensitivity and specificity of 94% and 83% for p16^INK4a^ IHC in identifying a transforming HPV infection in OPSCC [14]. Beyond these test-related parameters, what patients and clinicians are interested in knowing is as follows: What is the probability the tumor is truly HPV-driven if it is positive on the p16^INK4a^ IHC? This is the positive predictive value (PPV) and strictly depends on the prevalence of the disease being tested. Therefore, it is of paramount importance to know the current epidemiology of HPV-driven OPSCCs to estimate the reliability of p16^INK4a^ as a surrogate marker of HPV-driven carcinogenesis.

The aim of the present investigation was to evaluate the trend in the prevalence of HPV-driven OPSCC in the last 20 years in three centers in Northeastern Italy, with a focus on its impact on predictivity of p16^INK4a^ IHC in determining HPV-driven OPSCC.

## 2. Materials and Methods

### 2.1. Study Population

This is a multicenter retrospective study enrolling patients in three hospitals from two neighboring areas of Northeastern Italy (i.e., Veneto and Friuli Venezia Giulia regions), which share common lifestyle factors, including the prevalence of tobacco smoking and alcohol drinking [15,16]. Enrolling centers included the Treviso General Hospital, Treviso, in Veneto region, the joint Head and Neck Cancer unit of the Aviano National Cancer institute and the General Hospital “S. Maria degli Angeli”, Pordenone, and the Trieste University Hospital, Trieste, in Friuli Venezia Giulia region. The study included all consecutive patients that met the following inclusion criteria: (a) age ≥ 18 years; (b) diagnosis made between January 2000 and August 2022; (c) pathologically confirmed invasive SCC of the oropharynx (ICD-O-3 topography codes C01.9, C02.4, C05.1, C05.2, C09.0, C09.1, C09.8, C09.9, C10.0, C10.2, C.10.3, C10.8, and C10.9); and (d) available formalin-fixed paraffin-embedded (FFPE) specimens of the neoplastic lesion or availability of both high-risk HPV-DNA and p16^INK4a^ status. Data on 117 patients have been included in previously published report [17]. The whole study was approved by the ethic committees for clinical experimentation (CEP) of Treviso-Belluno provinces (Ethic votes: 345/AULSS9 and 421/AULSS9) and Friuli Venezia Giulia region (Ethic votes: CEUR-Os-041-ASUITS). Informed consent was obtained from all patients.

### 2.2. HPV-DNA Testing

DNA was extracted by the phenol–chloroform method or using a commercial Qiagen kit (Qiagen, Hilden, Germany), in accordance with the manufacturer’s instructions. Search and typing of HPV-DNA sequences were performed by PCR using MY09/MY11 primers and restriction fragment length polymorphism analysis of the amplified products, as described by Nobre et al. [18], or using the Linear Array HPV Detection and Genotyping Test (Roche Molecular Systems, Milan, Italy), as described by the manufacturer.

### 2.3. p16^INK4a^ Immunostaining

The expression of p16^INK4a^ protein was performed on FFPE sections by immunostaining using the primary antibody CINtec for V-kit (MTM laboratories, Heidelberg, Germany), or the BD Pharmingen IHC Detection kit, according to the manufacturer’s instructions. The expression results were scored as positive by using a 70% cut-point and considering the nuclear and cytoplasmic stain distribution. 

### 2.4. Statistical Analyses

A tumor was defined as HPV-driven when double positive for high-risk HPV-DNA and p16^INK4a^ overexpression. Patients with tumors positive for only one biomarker were included in the group of non-HPV-driven OPSCC. The prevalence of HPV-driven OPSCCs was calculated as the proportion of HPV-driven cases on the total number of OPSCC cases in the period; confidence intervals were calculated according to the Clopper–Pearson method. To evaluate the trends, four periods with similar number of cases were defined: 2000–2006, 2007–2012, 2013–2018, and 2019–2022. Analyses were further conducted in strata according to gender, age (i.e., <65 years vs. ≥65 years), and OPSCC subsite (i.e., tonsil and base of tongue vs. other subsites). Differences in prevalence across strata were evaluated through exact binomial test; the Kruskal–Wallis test was performed to evaluate differences across strata of continuous variables.

To evaluate the independent factors affecting the prevalence of HPV-driven OPSCCs, accounting for potential factor correlation, a multivariable logistic regression model was adopted. The odds ratio (OR) of being diagnosed with and HPV-driven OPSCC, with the corresponding 95% confidence interval (CI), was calculated, adjusting for potential predictors (namely, sex, age, cancer). 

PPVs and negative predictive values (NPVs) of p16^INK4a^ overexpression were calculated as a function of prevalence of HPV-driven OPSCC [19], ranging from 0% to 100%, using the pooled sensitivity and specificity estimated by Prigge et al. in their meta-analysis [14].

## 3. Results

A total of 390 consecutive patients with OPSCC—median age: 64 years, interquartile range: 57–71 years; 290 (74.4%) males—met the inclusion criteria and were included in the study (Table 1).

Overall, 125 cases (32.1%; 95% CI: 27.4–36.9%) were HPV-driven based on the double positivity for high-risk HPV-DNA and p16^INK4^a overexpression. Seventeen cases (4.5%; 95% CI: 2.6–6.9%) were p16^INK4a^ positive but high-risk HPV-DNA negative, while nine cases (2.3%; 95% CI: 1.1–4.3%) were high-risk HPV-DNA positive but p16^INK4a^ negative.

The prevalence of HPV-driven OPSCC was higher among women (42.0%) than among men (28.6%; *p* = 0.018), with a male to female prevalence ratio of 0.68. The median age at diagnosis was similar in patients with HPV-driven OPSCC and in those with non-HPV-driven cancers (median age: 64 vs. 63 years; *p* = 0.573). A statistically significant increase in the proportion of HPV-driven OPSCC was observed over time from 11.8% (95% CI: 5.6–21.3%) during 2000–2006 to 50.0% (95% CI: 39.8–60.2%) during 2019–2022 (*p* < 0.001).

HPV16 was detected in 114 cases (91.2%; 95% CI: 84.8–95.5%), HPV33 in 6 patients (4.8%; 95% CI: 1.8–10.2%), and HPV58 and HPV18 in two cases each; one case showed co-infection with HPV35 and HPV59. HPV16 was more frequently observed among younger patients (median age: 63 years, range 56–70 years), while other HPV-types were most prevalent in older subjects (median age: 78 years, range 69–84 years; *p* < 0.001).

The trend in the prevalence of HPV-driven OPSCC according to sex, age, and cancer subsite are shown in Figure 1. A statistically significant increase in the proportion of HPV-driven OPSCC was observed over time from 11.8% during 2000–2006 to 50.0% during 2019–2022. By stratifying for gender, the increasing trend of HPV-driven OPSCCs was evident in both genders, but was more marked in females, which increased from 5.3% in 2000–2006 to 65.4% in 2019–2022, compared to males (from 14.0% to 44.6% in the same periods). No differences emerged according to age at cancer diagnosis. The observed trend in prevalence was almost entirely dependent on the prevalence of SCCs of the tonsil and base of tongue, which increased from 13.6% to 59.0%.

The risk of being diagnosed with a HPV-driven OPSCC was lower in men (OR: 0.51; 95% CI: 0.30–0.87) and for cancer localized in the posterior pharyngeal wall/soft palate (OR: 0.06; 95% CI: 0.02–0.21; Table 2). Interestingly, the uprising trend with calendar period was confirmed (OR for 2019–2022 vs. 2000–2006: 9.37; 95% CI: 4.08–21.52), and this was independent from changes over time in distribution by sex and cancer subsite. 

By using the pooled sensitivity and specificity of p16^INK4a^ overexpression estimated by Prigge et al. in their meta-analysis [14], we drew a graph showing the variation of PPV and NPV of this surrogate marker of HPV-driven oncogenesis as a function of prevalence of HPV-driven OPSCC (Figure 2). Considering the observed prevalence of HPV-driven OPSCC, in our geographical area, the PPV of p16^INK4a^ overexpression is 89% for the tonsil and base of tongue SCC and only 29% for other subsites.

## 4. Discussion

The results of this investigation showed a progressive increase, from to 12% to 50%, in the fraction of OPSCCs attributable to a transforming HPV infection in a population of Northeastern Italy over a 20-year period. This increase concerned only cancers arising from the palatine tonsil and the base of tongue, coherently with the fact that these subsites are provided with the reticulated crypt epithelium, which is the main target of HPV-driven carcinogenesis [20]. Thus, the current prevalence of HPV-driven OPSCC was the highest ever recorded in Italy, which, similar to the other regions of Southern Europe, was counted among the regions with a relatively lower percentage of tumors attributable to HPV infection compared to the one observed in Northern Europe and Northern America [21]. Such a variation may certainly depend on the decrease in the fraction of OPSCCs attributable to other exposures, mainly tobacco smoking. This effect was already described in countries that experienced a decrease in the incidence of tobacco-related OPSCCs [22], as it is occurring in Italy for the most of tobacco-related cancers—including OPSCC—as a consequence of smoking restrictions [23]. Furthermore, the recent increase in the incidence of HPV-driven OPSCC is speculated to be a consequence of the maturation of the “baby boom” generation, experiencing the sexual revolution of the 1960s and 1970s [24]. Since the 1970s, there has been indeed a documented change in sexual behavior in several countries—including Italy—with a decrease in the age of first sexual intercourse, rise in the lifetime number of sexual partners, and increased oral sex [24,25,26], Which increase the spread of HPV infections.

Nonetheless, the overall increase in the incidence of oropharyngeal tumors observed in the last decades in our country [27,28] suggests that there is a real increase in the incidence of HPV-driven lesions. These results are consistent with recent trends observed in other European geographical areas including the Netherlands, France, Denmark, Sweden, and Southern Europe [2,29,30,31,32,33,34,35], with the growing prevalence of oral HPV infection due to changes in sexual habits likely being responsible for the observed epidemiological trends in oropharyngeal cancer [36]. 

However, the punctual prevalence can vary considerably according to the population and geographic area [32,33,34,35,37,38,39]. A different exposure to risk factors of oral HPV infection [40], as well as a different genetic susceptibility to persistent oral HPV infection [41], might partly explain these different figures. 

An interesting result confirming previous observations [17] is that non-HPV16 high-risk HPV types were found in OPSCC of subjects significantly older than subjects with HPV16-driven OPSCC. Among the high-risk strains, HPV16 has the greatest transforming properties [42] as well as being the most able to deregulate several innate immune-related pathways that block cytokine and chemokine production, antigen presentation, and adherence molecules [43]. Thus, the transformation process might take less time in case of HPV16 infection compared to an infection sustained by non-HPV16 high-risk strains and the clinical onset of the neoplastic disease could, therefore, occur earlier. Furthermore, given the high representation of lymphatic tissue in the oropharynx, there would be a more pronounced selective pressure towards HPV16 than in the cervix, which could explain the higher prevalence of HPV16-driven tumors in the oropharynx (91% of the HPV-driven cases in the present series) compared to what was observed in the cervical cancer (55%) [44].

Among the high-risk HPV non-HPV16 types, the most frequent was HPV33. This finding agrees with previous observations, which consistently show that HPV33 is the major cause of HPV-driven non-HPV16 oropharyngeal carcinogenesis [21,33]. This data appears interesting because the possibility has been raised that HPV33-driven HNSCC may not have a prognostic benefit over HPV-negative counterparts. In a retrospective analysis of The Cancer Genome Atlas HNSCC cohort, it was observed that HPV33-driven HNSCCs have a distinct genomic landscape characterized by a higher rate of aneuploidy, a peculiar immune microenvironment with reduced CD8 T-cell infiltration, and a worse OS compared to HPV16-driven tumors [45]. However, a recent cohort study did not confirm these findings and showed similar survival rates between patients with OPSCCs driven by HPV16 and HPV33 genotypes [9]. A large multicenter epidemiological study will, therefore, be necessary to clarify whether or not there are actually differences in the behavior of OPSCC induced by different high-risk HPV genotypes.

Another significant observation that emerged from the present analysis is that, in the last calendar period, the prevalence of HPV-driven OPSCC arising in subjects ≥65 years of age has equaled that of subjects <65 years of age; this indicates that in Northeastern Italy, HPV-driven OPSCC is no longer a disease of young individuals, as previously observed in the USA [41]. A cohort effect may explain this increase in the prevalence of HPV-driven OPSCC among older patients. Fortunately, previous investigations have confirmed that the survival benefit conferred by HPV is maintained, even in older subjects [46].

Although both HPV-driven and non-HPV-driven OPSCCs affected males more frequently than females, the relative proportion of HPV-driven cancers was higher in females (65% vs. 45% in the last calendar period). As also observed by other authors [47], this phenomenon is the consequence of a greater diffusion of smoking and alcohol habits in men, where the excess of OPSCCs due to exposure to these environmental factors dilutes the amount of HPV-driven OPSCCs. In 2021, in fact, smokers in Italy accounted for 23% of the men over 14 years of age, while 15% of women in the same age group were regular smokers [48].

Despite the TNM system having purely prognostic purposes and that it does not aim to suggest a particular therapeutic strategy, and that there are currently no specific treatments for HPV-driven OPSCC, it is known from both in vivo and in vitro observations that HPV-driven OPSCCs are more radiosensitive and chemo-sensitive [5,6]. Furthermore, for OPSCC, there are several possible choices of treatment rather than a treatment of choice. For instance, an advanced stage resectable tumor can be treated either with surgery followed by adjuvant (chemo)radiotherapy or with up-front chemoradiotherapy, reserving salvage surgery for non-responders. It is evident that in the face of an advanced HPV-driven OPSCC, even if it is resectable, one will be more optimistic in undertaking a non-surgical treatment strategy compared to a carcinoma of the same stage but HPV-negative. In fact, the NCCN guidelines recommend concomitant chemoradiotherapy for T4 or N3 p16^INK4a^-positive OPSCCs [49]. For the overall prevalence of HPV-driven OPSCC of 80%, 50%, and 20%, the PPV of p16^INK4a^ overexpression would be 96%, 85%, and 58%, respectively. Furthermore, considering the low prevalence of HPV-driven SCC in these subsites, it has almost no validity for SCCs originating from soft palate and posterior oropharyngeal wall. We, therefore, believe that each Institution should know the epidemiological landscape of HPV-driven oropharyngeal carcinogenesis in its own geographic area in order to establish a more reliable diagnostic approach toward OPSCCs. Furthermore, the possibility, which in any case needs further confirmation, that OPSCC driven by non-HPV16 high-risk HPV genotypes may not have a prognostic benefit compared to HPV-negative counterpart [45], underlines how it could become important not only to search for DNA sequences for high-risk HPV but also to perform HPV genotyping. Thus, when the PPV of p16^INK4a^ is low, we strongly recommend defining an oropharyngeal cancer as HPV-driven when double positive for high-risk HPV-DNA sequences and p16^INK4a^. Indeed, other biomarkers highly indicative of a transforming HPV infection, such as E6 and E7 mRNA detection, are not easily usable in clinical practice [11].

Finally, in light of this progressive increase in HPV-driven OPSCC, health policies should actively promote vaccination against HPV in both females and males. HPV vaccines have been shown to be effective in reducing the prevalence of oral and oropharyngeal HPV infections, with an average relative prevention percentage of 83% [50]. In 2020, HPV recombinant 9-valent vaccine received Food and Drug Administration approval for an expanded indication both in females and males to include the prevention of OPSCC caused by HPV types 16, 18, 31, 33, 45, 52, and 58 [48]. We believe that this indication should be introduced by the competent authorities also in our country.

Some potential limitations have to be acknowledged. A first limitation is the relatively small number of patients included in our cohort. Secondly, this is a retrospective study based on clinical data from participating centers and, different from population-based studies, it may suffer from selection bias. However, the participant hospitals are referral centers for OPSCC in their catchment area, and patients’ health mobility is negligible. Furthermore, FFPE specimens were obtained based on availability within pathology archives, and this limited the sample size, particularly in the early study period. Finally, prospectives are necessary to better clarify the accuracy of p16^INK4a^ overexpression as a surrogate marker of transforming HPV infection. The strengths of our study are the use of two biomarkers for HPV-driven determination and the long period of patient enrolment.

## 5. Conclusions

The prevalence of HPV-driven tonsils and base of the tongue SCCs continued to increase, even in the most recent time period with a more marked increase being observed in females compare to males. Consequently, it is urgent to establish politics to increase the information of the Italian population on this aspect. When using p16^INK4a^ overexpression as a surrogate marker of transforming HPV infection, each institution should consider their own subsites-specific prevalence rates of HPV-driven OPSCC as these significantly impact on its positive predictive value.

## Figures and Tables

**Figure 1 cancers-15-02643-f001:**
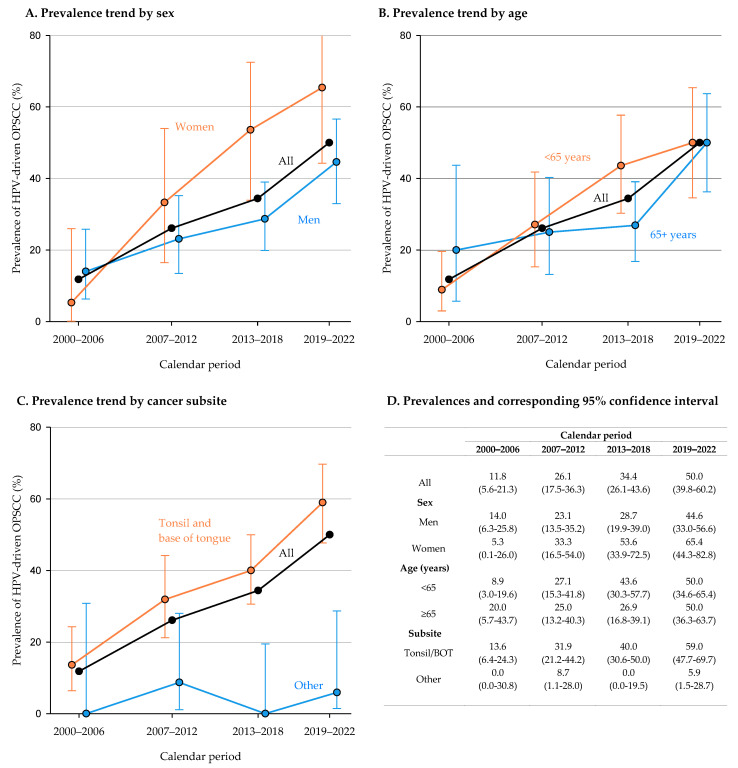
Trend in the prevalence of HPV-driven oropharyngeal squamous cell carcinoma (OPSCC) according to sex (**A**,**D**), age (**B**,**D**), and cancer subsite (**C**,**D**).

**Figure 2 cancers-15-02643-f002:**
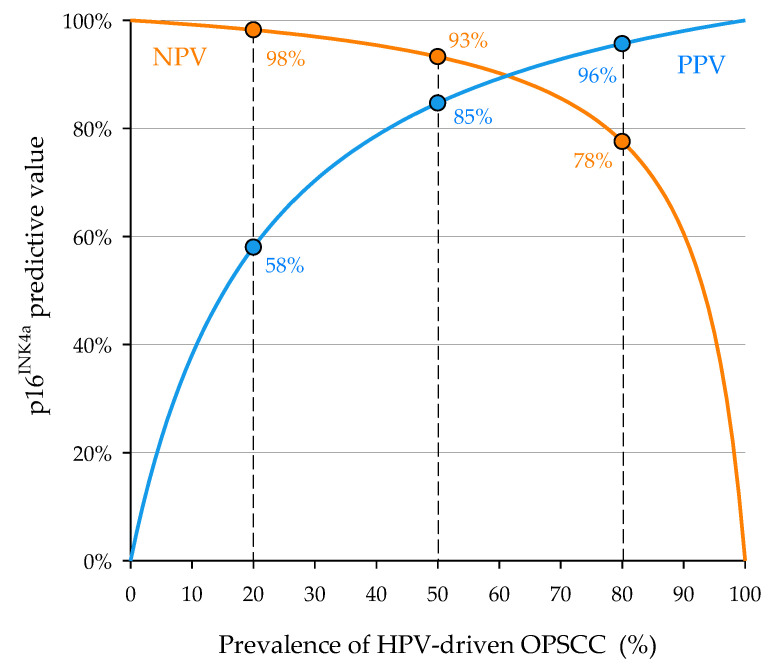
Positive (PPV) and negative predictive values (NPV) of p16^INK4a^ overexpression as a function of prevalence of HPV-driven oropharyngeal squamous cell carcinoma (OPSCC), calculated using pooled sensitivity and specificity estimated by Prigge et al. [14].

**Table 1 cancers-15-02643-t001:** Demographic and Clinical Characteristics of the Study Cohort.

	All		HPV-Driven ^1^ OPSCC			Fisher’s Exact Test
	No.	%	No	%	(95% CI)	
**All patients**	390		125	32.1	(27.4–36.9)	
**Age, median (IQR), year**	64	(57–71)		64	(57–71)	
**Sex**						
Female	100	25.6	42	42.0	(32.2–52.3)	*p* = 0.018
Male	290	74.4	83	28.6	(23.5–34.2)	
**Subsite**						
Tonsil	228	58.5	89	39.0	(32.7–45.7)	*p* < 0.001
Base of tongue	95	24.4	33	34.7	(25.3–45.2)	
Posterior pharyngeal wall	25	6.4	1	4.0	(0.1–20.4)	
Soft palate	42	10.8	2	4.8	(0.6–16.2)	
**Overall stage (UICC/AJCC 8th)**						
I	62	15.9	33	53.2	(40.1–66.0)	*p* < 0.001
II	76	19.5	52	68.4	(56.7–78.6)	
III	70	18.0	34	48.6	(36.4–60.8)	
IV	177	45.4	5	2.8	(0.9–6.5)	
Unknown	5	1.3	1	20.0	(0.5–71.6)	
**Calendar period**						
2000–2006	76	19.5	9	11.8	(5.6–21.3)	*p* < 0.001
2007–2012	92	23.6	24	26.1	(17.5–36.3)	
2013–2018	122	31.3	42	34.4	(26.1–43.6)	
2019–2022	100	25.6	50	50.0	(39.8–60.2)	
**HPV type**						
HPV16			114	91.2	(84.8–95.5)	
HPV33			6	4.8	(1.8–10.2)	
HPV18			2	1.6	(0.2–5.7)	
HPV58			2	1.6	(0.2–5.7)	
HPV35+ HPV 59			1	0.8	(0.0–4.4)	

AJCC: American Joint Committee on Cancer; CI: confidence interval; HPV: human papillomavirus; IQR: interquartile range; OPSCC: oropharyngeal squamous cell carcinoma; UICC: Union for International Cancer Control. ^1^ Based on double positive for high-risk HPV-DNA and p16^INK4a^ overexpression.

**Table 2 cancers-15-02643-t002:** Risk of being diagnosed with an HPV-driven oropharyngeal squamous cell carcinomas according to potential predictors.

	Univariate Model ^1^		Multivariable Model ^2^	
	OR	(95% CI)	OR	(95% CI)
**Age, year** (Reference: <55)				
55–64	0.93	(0.51–1.71)		
65–74	0.96	(0.53–1.75)		
≥75	1.13	(0.54–2.35)		
**Sex** (Reference: Woman)				
Man	0.55	(0.35–0.89)	0.51	(0.30–0.87)
**Subsite** (Reference: Tonsil)				
Base of tongue	0.83	(0.51–1.37)	0.82	(0.48–1.40)
PPW/Soft palate	0.07	(0.02–0.24)	0.06	(0.02–0.21)
**Calendar period** (Reference: 2000–2006)				
2007–2012	2.63	(1.14–6.07)	3.29	(1.39–7.80)
2013–2018	3.91	(1.77–8.60)	4.48	(1.98–10.13)
2019–2022	7.44	(3.35–16.54)	9.37	(4.08–21.52)

CI: confidence interval; OR: odds ratio; PPW: posterior pharyngeal wall. ^1^ Estimated from unconditional logistic regression model. ^2^ Estimated from unconditional logistic regression model, adjusting for sex, subsite, and calendar period.

## Data Availability

Data are available for research purpose upon reasonable request to Paolo Boscolo-Rizzo.

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
