# Peer review of "Rising Trend in the Prevalence of HPV-Driven Oropharyngeal Squamous Cell Carcinoma during 2000–2022 in Northeastern Italy: Implication for Using p16INK4a as a Surrogate Marker for HPV-Driven Carcinogenesis"

_cancers, 2023, doi:10.3390/cancers15092643_

Round 1

Reviewer 1 Report (Previous Reviewer 1)

I suggest this study to be accepted.

English is overall good, no major issues detected.

Reviewer 2 Report (New Reviewer)

The prevalence  of oropharyngeal squamous cell carcinomas (OPSCCs) related to HPV infection are increasing all over the world. OSCC is very significant problem both for epidemiological and clinical point of few. The Authors evaluated the trend in the prevalence of HPV-associated OPSCC in the last 20 years in three centers in northeastern Italy. This is a retrospective study. Clinical specimens were formalin-fixed paraffin- embedded specimens. Both high-risk HPV- DNA and p16INK4a status have been detected. In 91.2% of cases was detected type HPV 16. It is interesting to see such a significant increase in frequency of base of  tongue increased from 13.6% to 59.0%.which is probably related to smoking.it would be worth analyzing the correlation with other risk factors and G feature and TNM classification.How the clinical stage of this cancer has changed over time.

This manuscript is a resubmission of an earlier submission. The following is a list of the peer review reports and author responses from that submission.

Round 1

Reviewer 1 Report

In this retrospective study, the authors used a fair amount of clinical patients’ tumor samples and information to do a very comprehensive analysis. They recalculated the predictive value (both positive and negative) of p16INK4a overexpression based on the prevalence of HPV-driven OPSCCs, they also discovered the rising trend of HPV-driven OPSCCs in northeastern Italy specifically, but mainly just in tonsil and base of tongue. The statistical analysis is convincing and extensive. The study is of significant clinical relevance.

The current study is decent and rigorous. However, the size of content is not enough to be published on a journal like Cancers. The conclusion is more like a suggestion rather than summary of solid results, further investigation or experiment is required. For example: 1. What could be the possible factors that impact the prevalence uprising? More correlation analysis can be done. 2. When the predictive value is low, are there any other alternative surrogate markers that can be used? 3. Not just retrospective study, prospective study can also be done to verify the accuracy of p16INK4a. 4. Any other data available from other areas? Etc.

Reviewer 2 Report

In my opinion this is an interesting, well written and methodologically well done work that demonstrates an increase in the prevalence of oropharyngeal carcinomas etiologically related to high risk HPV in some regions of Northern Italy, in the last decades. The main cause is the change of sexual habits in today's world, promiscuity and lack of preventive measures. All this suggests, among other things, the need to establish politics to increase the information of the Italian population on this aspect. I believe that this last point should be included in the conclusions of the study to give a wake-up call to the health authorities of this country. Except for this comment, I believe that this paper should be published in its current format.

Reviewer 3 Report

Dear authors of article

"Rising Trend in Prevalence of HPV-driven Oropharyngeal Squamous Cell Carcinoma during 2000-2022 in Northeastern Italy: Implication for Using p16INK4a as Surrogate Marker for HPV-driven Carcinogenesis"

This paper presents the (almost) same results that were published 2 years earlier: Del Mistro A, Frayle H, Menegaldo A, Favaretto N, Gori S, Nicolai P, Spinato G, Romeo S, Tirelli G, da Mosto MC, Polesel J, Boscolo Rizzo P. Age-independent increasing prevalence of Human Papillomavirus-driven oropharyngeal carcinomas in North-East Italy. Sci Rep. 2020 Jun 9;10(1):9320. doi: 10.1038/s41598-020-66323-z. PMID: 32518378; PMCID: PMC7283341.

I would also like to point out that the article misunderstands the value of PPV. PPV has nothing to do with disease incidence and trend. But, the PPV tells you if the test correctly detects the disease. The test's prognosis (whether it's good) is affected by its sensitivity and specificity, which affects how well it detects disease (PPV) or excludes it (NPV).